# Malaria prevalence, knowledge and associated factors among household heads in Maygaba town, Ethiopia

**Tsegay Gebremaryam Yhdego[1], Asnake Desalegn Gardew[2], Fitsum Tigu Yifat[2]***

**1** Department of Biology, College of Natural and Computational Sciences, Addis Ababa University, Addis Ababa, Ethiopia, **2** Department of Microbial, Cellular and Molecular Biology, College of Natural and Computational Sciences, Addis Ababa University, Addis Ababa, Ethiopia

\* fitsum.tigu@aau.edu.et

## Abstract

Malaria remains a leading public health issue in Ethiopia, despite wide use of insecticide-treated mosquito nets (ITNs). Consistent use of ITNs reduces malaria transmission by 90%. However, coverage and proper use of ITNs are the major challenge for most rural settings of Ethiopia. We assessed the prevalence of malaria, ITNs use and associated factors among household heads in Maygaba town, northwest Ethiopia. A cross-sectional study was carried out among 403 households using a structured questionnaire. Blood samples were collected from household heads and examined for malaria parasites by microscopy. Chi-square test and logistic regression were used to check the association between the dependent and independent variables. Of the 403 blood samples, 19 (4.7%) had malaria parasites (12 cases of *Plasmodium falciparum* and 7 cases of *Plasmodium vivax*). Majority of the respondents were aware of the cause, communicability, preventability and curability of the disease. ITNs use was identified as the main malaria prevention method. About 64% (257) of the respondents owned ITNs, of which, 66.9% (172) consistently slept under the nets during the peak malaria season. Majority of the respondents (83.6%) had positive attitude towards the benefit of sleeping under ITNs. Educational status and livelihood of the respondents had statistically significant ($p < 0.05$) association with malaria knowledge. Malaria infection was significantly ($p < 0.05$) higher among household heads aged 18–30 years (AOR = 5.982; 95% CI = 1.715–20.863). Moderate prevalence rate and acceptable ITNs ownership was detected among the community of Maygaba town. However, a considerable proportion had misconceptions about the use of ITNs. A comprehensive malaria control measures and health education about the use of ITNs should be in place for effective prevention and control of malaria in the locality.

## Introduction

Malaria is an infectious protozoal disease transmitted by the bite of a female *Anopheles* mosquitoes. The parasitic agents associated with malaria in humans belong to the genus

**Data Availability Statement:** All relevant data are within the paper.

**Funding:** The author(s) received no specific funding for this work.

**Competing interests:** The authors declare that they have no competing interests.

*Plasmodium*, with five species, namely *P. falciparum*, *P. vivax*, *P. malariae*, *P. ovale* and *P. knowlesi* [1, 2]. In Ethiopia, *P. falciparum* and *P. vivax* are the most common and widely distributed species that account for 60% and 40% of malaria cases, respectively [3]. Among several Anopheline species that transmit malaria in humans, only *Anopheles arabiensis* is recognized as primary vector in Ethiopia, while others *Anopheles pharoensis*, *Anopheles funestus* and *Anopheles nili* are considered as the secondary vectors [4].

Ethiopia is one of malaria prone countries in Africa with 52.7 million people at risk of malaria infection [5, 6]. This is due to the presence of favorable conditions for vector development and multiplication [7, 8]. In connection with this, the disease is mainly seasonal with variable transmission rate across different agroecology and usually coincides with the peak agricultural activities that would greatly affect the socio-economic development of the country [9]. In the country, the overall prevalence of malaria reached 37.6% in the mixed regions of lowlands and midlands followed by 20.7% reported in the lowlands [9].

Several reports indicated that consistent and proper utilization of ITNs, particularly the long-lasting insecticidal nets (LLINs) have been shown to be effective in reducing malaria transmission by 90% [10–12]. Although the Federal Democratic Republic of Ethiopia Ministry of Health (FMOH) reported that most households had at least two ITNs per family [13], there are still knowledge gaps about consistent and correct use of ITNs by most communities of Ethiopia [10]. Additionally, full coverage and proper utilization of ITNs are vital for the prevention and control of malaria [14], however, there are problems associated with sustainable distribution and timely replacement of the old ITNs, seasonality of malaria, and lack of knowledge about the disease, ITNs and the vector. A recent study also revealed low coverage and improper use of ITNs among many households [7]. Furthermore, the coverage and utilization of ITNs also differed between regions [15].

The main determinants in the ownership and utilization of ITNs in sub-Saharan African as reviewed by Singh et al. [16] are educational level, knowledge of malaria, socio-economic status and parity and community participations. These factors contribute to the low efficacy (60%) of the available ITNs [8, 16, 17]. In northern Ethiopia, including the study area, out of the total malaria exposed households, only 74% of them received one ITN [18]. Besides the ITNs distribution, regular insecticide residual spraying (IRS) has been practiced in the study area. However, till now no study has been conducted in the study area about the current status of malaria and associated factors. Therefore, the study aimed to assess the prevalence of malaria, ITNs use and associated factors among household heads in Maygaba town, northwest Ethiopia.

## Methods

### Study area and population

A community-based cross-sectional study was conducted in Maygaba town of Welkait district, northwest Ethiopia, from March to May 2019. The town is located at 929 Km from the capital city-Addis Ababa. It has six rural villages with an estimated 7,039 households. Each village has an average family size of 4.4 persons per household and a total population of 30,974; of which 15,642 are male and 15,332 are female. Three villages (Korarit, Maygaba and Adijamus) with a total population of 16,781 in 3,814 households were involved in the study. Male and female dwellers in these three villages were 8,474 (50.5%) and 8,307 (49.5%), respectively. The population size of each selected village was 7,266 (43.3%) in Korarit, 4,962 (29.6%) in Maygaba and 4,553 (27.1%) in Adijamus. On average, each village had 1,271 households and 5,593 population. The study area is characterized by altitudinal ranges of 677 to 2,755 meters above sea

level. Its annual temperature and unimodal rainfall distribution are 17.5–25°C and 700–1800 mm, respectively.

Persons eligible for the parasitological survey were heads of the household or their designate willing to give blood samples for malaria testing and no history of anti-malaria therapy within the previous two weeks. Thus, only family members older than 18 years, who gave blood samples for malaria diagnosis, and volunteered for interviews were included in KAP and parasitological surveys. Family members who were not present at home during the study period, unable to communicate and mentally handicapped were excluded from the KAP study.

## Sample size and sampling procedure

The sample size was calculated following single population proportion formula, $n = \left(\frac{z\alpha}{2}\right)^2 \times \frac{p(1-p)}{d^2}$ reported elsewhere [19]. Assuming that half of the respondents had knowledge on cause and transmission of malaria and use of ITNs with an estimated malaria prevalence rate of 50% ($p = 0.5$) at 95% confidence interval ($Z\alpha/2 = 1.96$) and 5% of marginal error ($d = 0.05$). Based on the formula, calculated value of 384 plus 10% non-response rate, the total participants were 422.

A multi-stage random sampling technique was implemented (Fig 1). A total of 422 households were selected from the 12 *Gujiles* (health development group or cluster, used to address the rural people with health packages). On average each village contains about 28 *Gujiles* and each *Gujile* also contains 45 households. The proportion of households for each *Gujile* was determined by dividing the total sample size (422) by the total selected *Gujiles* (12) and about 35 households per *Gujile* was included in the study. This procedure was used because *Gujiles* are constructed from approximately equal numbers of households. A systematic random sampling technique was used to select every n[th] household from each *Gujile*. When possible, the head of the household was enrolled for KAP, in the absence of a household head, any family member older than 18 years and willing to participate in the KAP as well as parasitological studies were included.

## Data collection

**Cross-sectional study.**    Qualitative data was collected by a face-to-face interview using a structured questionnaire. The questionnaire contained both close- and open-ended questions about KAP on malaria and use of ITNs and prepared from previous studies associated with malaria [20–22]. The questionnaire was developed in English, translated into both Amharic and Tigrigna (local spoken languages) and checked for correctness of the translation by fluent speakers of both languages. Participants who were unable to read and write were assisted by an enumerator (who has an ability to speak Amharic and Tigrigna languages). The questionnaire was pre-tested by a preliminary survey in some *Gujiles* outside the sampled area and the content, completeness and suitability towards the target study was validated. Finally, the questionnaire was amended to suit the objectives of the current study.

Generally, it contained 28 questions under three categories; 1) socio-demographic data: sex, age, level of education, livelihood of the household, family size, livestock ownership and type of house; 2) basic knowledge about the cause of malaria, transmission and ways to prevent malaria and use of ITNs to prevent malaria and their treatment seeking behavior of the respondent; 3) basic awareness and behaviors towards the use of ITNs: including households' awareness and source of information about ITNs, ownership and number of ITNs in a family, reasons for non-possession. Additionally, there were questions that explored respondents' frequency of ITNs use, the benefits of ITNs, and which family members are given priority to use ITNs as well as their experience of ITNs re-treatment with insecticide.

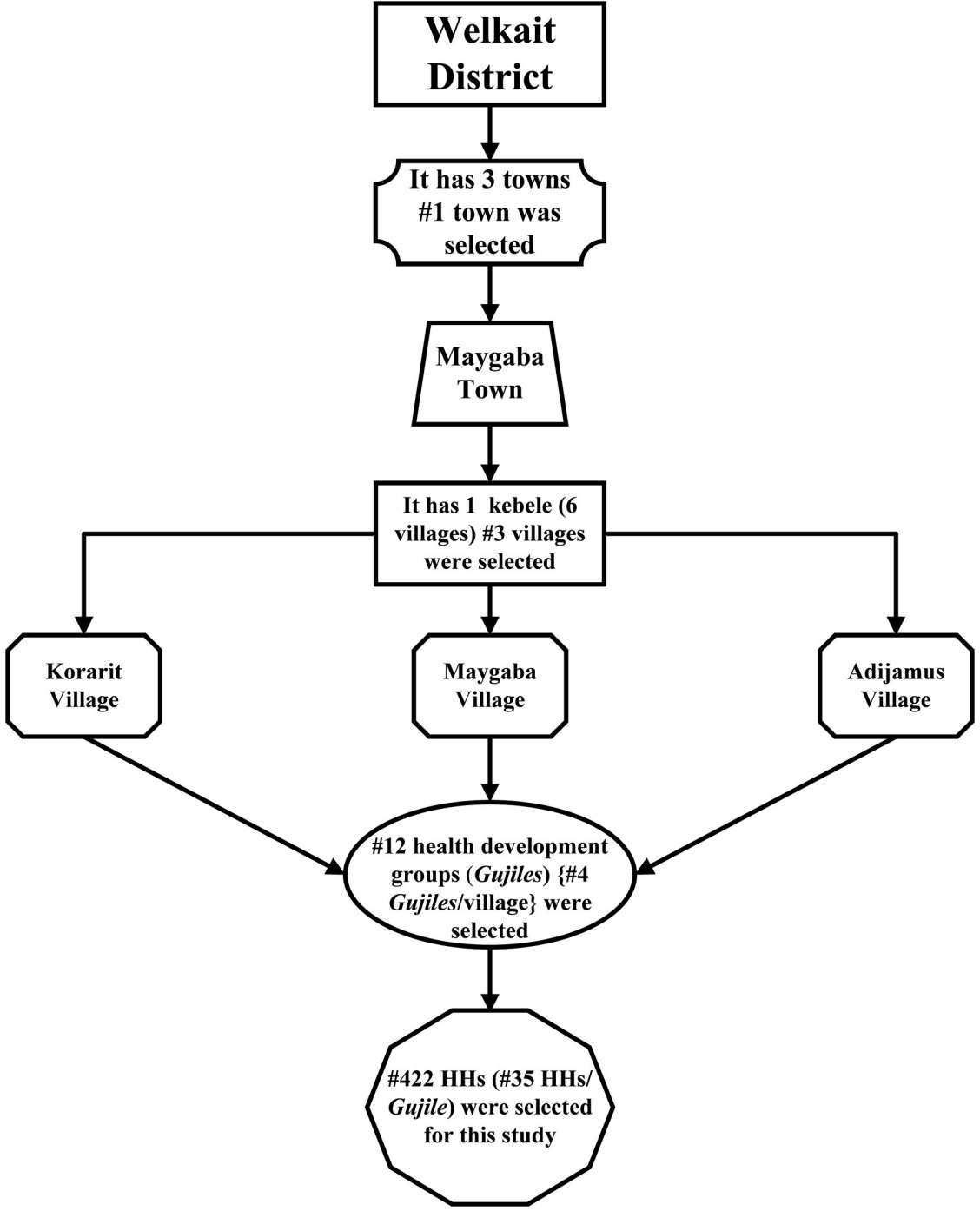

**Fig 1. Flow chart indicating the multi-stage cluster random sampling technique.**

**Parasitological examination.** Finger-prick blood samples were collected by laboratory technicians. Thin and thick films were made on the slide and then properly labeled. The thin films were fixed with pure methanol at the site. After air drying in an upright position, all the slides were placed in the slide box and carefully transported to the Maygaba health center (MHC) for parasitological examination. Both the thick and thin films were stained with

Giemsa (3%) solution as per the standard microscopic protocol [23]. The presence of malaria parasites in 100 fields were examined from the thick films. The thin films were used for identification of *Plasmodium* species. In order to control the quality of slide reading, a second microscopist was employed to re-examine all the slides for parasite identification. Finally, about 5% of the slides were randomly selected and re-examined by the senior expert (the third reference reader) at MHC for quality control. No discrepancy was identified among the readers.

## Outcome measure

The outcome variables in this study were knowledge about malaria and malaria infection. All the knowledge related variables were recorded to binary level that means for correct response was coded 1 while an incorrect response was coded 0. Those questions having multiple correct responses, the same scoring was followed but the score was divided by the total number of multiple responses in the question to normalize the result between 0 and 1. The total scores were then standardized to the range of 0 and 1. Finally, the mean score was calculated and tertiles of the composite score were used as the cut-off to group the knowledge levels as poor (score $\leq 0.4$), good (score $> 0.4$–0.7), and high (score $> 0.7$–1) following the method reported elsewhere with modification [24].

## Data analysis

Completeness and consistency of the data were checked and double entered into SPSS version 22.0 software (SPSS Inc, Chicago, IL, USA) for statistical analysis. Descriptive statistics was used to tabulate and describe the cross-sectional data. Chi-square ($\chi^2$) test and logistic regression were used to determine the association between the dependent and independent variables. Statistical significance was defined at $p < 0.05$.

## Ethics statement

The study was ethically approved by the Health Bureau of Welkait district and College of Natural and Computational Sciences, Institutional Review Board (Ref. No. CNSDO/185/12/19), Addis Ababa University. Before conducting the investigation, the researcher discussed the study with local administrative bodies in the study area. All the respondents were clearly informed about the purpose of the study and kindly asked to participate in the study. Prior to administering the questionnaire and blood sample collection, all participants agreed and signed the consent form. Blood samples were collected by trained staff of MHC and all malaria positive cases were carefully treated according to the national malaria guidelines [25].

## Results

### Participant demographics

A total of 422 households were surveyed, but data from 19 households were excluded from further analysis due to incomplete data. Four hundred three participants' information were compiled and had a response rate of 95.5%. The details of the participants' demographic data were summarized in Table 1. Majority of the respondents were male (61.5%, N = 248) and almost half (48.9%, *N* = 197) of the respondents were in the age category of 18–30 years followed by 31–40 years age group accounted 37.2% (*N* = 150). Most of the households had greater than five person per household (62.3%, *N* = 251); over half (53.6 and 54.3%) of the respondents were illiterate and farmers, respectively.

**Table 1. Socio-demographic characteristics of the respondents (N = 403).**

| Variables | Category | Frequency (%) |
|---|---|---|
| Sex | Male | 248 (61.5) |
| | Female | 155 (38.5) |
| Age | 18–30 | 197 (48.9) |
| | 31–40 | 150 (37.2) |
| | >41 | 56 (13.9) |
| Education | Illiterate | 216 (53.6) |
| | Elementary | 173 (42.9) |
| | Secondary | 14 (3.5) |
| Livelihood | Farmer | 219 (54.3) |
| | Merchant | 10 (2.5) |
| | Student | 114 (28.3) |
| | Housewife | 57 (14.1) |
| | Daily laborer | 3 (0.7) |
| Family size (age <18 years) | 1–3 persons | 46 (11.4) |
| | 4–5 persons | 106 (26.3) |
| | > 5 persons | 251 (62.3) |
| Livestock ownership | Yes | 349 (86.6) |
| | No | 54 (13.4) |
| Ownership & type of house* | Modern | 231 (57.3) |
| | Conventional | 172 (42.7) |

* Modern house: made-up from basalt foundation, thick stone wall, windows, doors, lintels, floors and ceilings are carved out of wood and a conical metal roof; Conventional house: made-up from the abundant stone, mud, tree branches, timber or wood, soil or earth and slate.

## Prevalence of malaria

The overall prevalence of malaria in the study area was 4.7% with an asymptomatic infection category of twelve cases of *P. falciparum* and seven cases of *P. vivax* (Table 2).

## Awareness about cause, transmission, prevention and control of malaria

Majority of the participants (91.3%, *N* = 368) living in the study area had awareness about the presence of malaria in their vicinity. Similarly, most of the respondents (78.4%, *N* = 316) replied that malaria is a communicable disease, of which (75.6%, *N* = 239) replied that malaria is transmitted by the bite of a female *Anopheles* mosquitoes. Of the 239 respondents, 207 (86.6%) replied that night time is suitable for mosquito biting. Most (82.4%, *N* = 332) of respondents replied that malaria is a preventable disease. Among the three-hundred and thirty-two respondents, the majority (81.0%, *N* = 269) of participants replied that ITNs is the main protection tool for malaria infection. Likewise, a large proportion (84.1%, *N* = 339) of the respondents recognized that malaria is treatable if managed earlier, of whom (82.3%, *N* = 279) of them preferred health centers for treatment (Table 3).

## Use of ITNs

According to the survey, the majority of the respondents (75.7%, *N* = 305) had heard about ITNs as it summarized in Table 4. Although their sources of information varied, most (73.9%, *N* = 227) of them were informed by health workers. With regard to ITNs ownership, about

**Table 2. Demographic characteristics of the respondents diagnosed for malaria at MHC (*N* = 403).**

| Variables | Category | No tested (%) | Slide-positive (%) | *P. falciparum* | *P. vivax* |
|---|---|---|---|---|---|
| Sex | Male | 248 (61.5) | 12 (3.0) | 8 | 4 |
| | Female | 155 (38.5) | 7 (1.7) | 4 | 3 |
| | Total | 403 (100) | 19 (4.7) | 12 | 7 |
| Age (year) | 18–30 (male) | 126 (31.3) | 10 (2.5) | 6 | 4 |
| | 18–30 (female) | 71 (17.6) | 6 (1.5) | 4 | 2 |
| | 31–40 (male) | 80 (19.8) | 1 (0.3) | 1 | 0 |
| | 31–40 (female) | 70 (17.4) | 1 (0.3) | 0 | 1 |
| | > 41 (male) | 42 (10.4) | 1 (0.3) | 1 | 0 |
| | > 41 (female) | 14 (3.5) | 0 (0.0) | 0 | 0 |
| | Total | 403 (100) | 19 (4.7) | 12 (63.2%) | 7 (36.8%) |

64.0% (*N* = 257) of the respondents possessed ITNs. Of the two-hundred and fifty-seven ITNs owners, 71.6% (*N* = 184) of the respondents had one ITN per family, followed by two (15.2%, *N* = 39) and three (13.2%, *N* = 34). On the other hand, from one-hundred and forty-six non-owners, 56.2% (*N* = 82) mentioned that they didn't receive ITN, while 43.8% (*N* = 64) replied that their ITN was worn-out due to wear and tear. Beside the good knowledge of respondents

**Table 3. Awareness about cause, transmission, prevention and control of malaria (*N* = 403).**

| Variables | Category | Frequency (%) |
|---|---|---|
| Ever heard about malaria (awareness) | Yes | 368 (91.3) |
| | No | 11 (2.7) |
| | I do not know | 24 (6.0) |
| Malaria is a communicable disease (knowledge) | Yes | 316 (78.4) |
| | No | 64 (15.9) |
| | I do not know | 23 (5.7) |
| Cause of malaria (n = 316) (knowledge) | Mosquito bite | 239 (75.6) |
| | Environmental change | 44 (13.9) |
| | Lack of environmental sanitation | 33 (10.4) |
| Time of mosquitos bite mostly (n = 239) (knowledge) | Day | 15 (6.3) |
| | Night | 207 (86.6) |
| | Any time | 17 (7.1) |
| Malaria is preventable (knowledge) | Yes | 332 (82.4) |
| | No | 24 (6.0) |
| | I do not know | 47 (11.7) |
| Prevention methods (n = 332) (knowledge as well as practice) | Apply environmental sanitation | 9 (2.7) |
| | Use of ITN | 269 (81.0) |
| | Fumigation | 14 (4.2) |
| | Wearing long sleeved clothes | 40 (12.1) |
| Malaria is treatable (practice) | Yes | 339 (84.1) |
| | No | 21 (5.2) |
| | I do not know | 43 (10.7) |
| Ways of treatment (n = 339) (practice) | Traditional healer | 22 (6.5) |
| | Health center | 279 (82.3) |
| | Buy drug from pharmacy | 30 (8.9) |
| | Others | 8 (2.4) |

**Table 4. The use of ITNs among the respondents (N = 403).**

| Variables | Category | Frequency (%) |
|---|---|---|
| Ever heard about ITN | Yes | 305 (75.7) |
| | No | 98 (24.3) |
| Source of information about ITN (n = 305) | Mass media (TV & radio) | 65 (21.3) |
| | Health workers | 227 (74.4) |
| | Local leader | 13 (4.3) |
| ITN possessed | Yes | 257 (63.8) |
| | No | 146 (36.2) |
| ITN per family (n = 257) | One | 184 (71.6) |
| | Two | 39 (15.2) |
| | Three | 34 (13.2) |
| Reason for ITN non-possession (n = 146) | Worn-out | 64 (43.8) |
| | Not received | 82 (56.2) |
| Duration of ITN utilization (n = 257) | 1–2 years | 103 (40.1) |
| | >2 years | 154 (59.9) |
| Sleeping under ITN is beneficial | Yes | 337 (83.6) |
| | No | 66 (16.4) |
| What are the benefits of ITN (n = 337) | Protect mosquito bite | 292 (86.6) |
| | Comfortable sleep | 35 (10.4) |
| | Others | 10 (3.0) |
| ITN used last night (n = 257) | Yes | 173 (67.3) |
| | No | 84 (32.7) |
| Frequency of ITN use (n = 257) | Regularly | 73 (28.4) |
| | Malaria season | 172 (66.9) |
| | Sometimes | 12 (4.7) |
| Who mostly used ITN in the family (n = 257) | Mother and children (<5) | 102 (39.7) |
| | Pregnant woman | 155 (60.3) |
| Re-treated ITN with insecticide (n = 257) | Yes | 67 (26.1) |
| | No | 190 (73.9) |
| If no, what are the reasons (n = 190) | Lack of awareness | 89 (46.8) |
| | Lack of insecticide | 101 (53.2) |

towards ITNs utilization, most of the respondents (83.6%, $N = 337$) also had a positive attitude towards the benefits of sleeping under ITN. Out of 83.6% ($N = 337$) respondents, 87% ($N = 292$) of them believed that ITN can be used to protect from mosquito bite. While the rest of the respondents used the ITN for purposes other than malaria prevention such as comfortable sleeping and others.

Out of the two-hundred and fifty-seven respondents who had ITNs, a considerable number (67.3%, $N = 173$) of respondents had slept under ITN last night at the time of interview. A similar figure (60%, $N = 154$) had the best experience of utilizing their ITN over 2 years and 40% ($N = 103$) of them used it for 1.5 years on average. Nearly 67% ($N = 172$) of the respondents commonly used their ITN during the peak season of malaria. Mostly, pregnant women had given the priority to use ITN among the family members (60.3%, $N = 155$), followed by mother and children under five (39.7%, $N = 102$). Despite the possession of ITN, most of the respondents (74%, $N = 190$) did not re-treat their ITNs with insecticide due to two main reasons; lack of awareness and lack of insecticide. However, a quarter of the households (26%, $N = 67$) properly utilized their ITNs with insecticide treatment as described in Table 4.

**Table 5. Association of knowledge about malaria with selected socio-demographic characteristics.**

| Variable | Knowledge about Malaria | | | $\chi^2$ | p-value |
|---|---|---|---|---|---|
| | Poor (F, %) | Good (F, %) | High (F, %) | | |
| Sex | | | | | |
| Male | 75 (30.2%) | 98 (39.5%) | 75 (30.2%) | 0.026 | 0.871 |
| Female | 47 (30.3%) | 63 (40.7%) | 45 (29%) | | |
| Age | | | | | |
| 18–30 | 58 (29.4%) | 81 (41.1%) | 58 (29.4%) | 0.851 | 0.654 |
| 31–40 | 44 (29.3%) | 62 (41.3%) | 44 (29.3%) | | |
| > 41 | 21 (37.5%) | 22 (39.3%) | 13 (23.2%) | | |
| Family size | | | | | |
| 1–3 person | 15 (32.6%) | 18 (39.1%) | 13 (28.3%) | 3.861 | 0.145 |
| 4–5 person | 36 (34%) | 42 (39.6%) | 28 (26.4%) | | |
| > 5 person | 71 (28.3%) | 100 (39.8%) | 80 (31.9%) | | |
| Livelihood | | | | | |
| Farmer | 69 (31.5%) | 86 (39.3%) | 64 (29.2%) | 6.219 | 0.013* |
| Non-Farmer | 52 (28.3%) | 85 (46.2%) | 47 (25.5%) | | |
| Education | | | | | |
| Illiteracy | 75 (34.6%) | 87 (40.3%) | 54 (25%) | 8.201 | 0.006* |
| Elementary & above | 47 (25.1%) | 66 (35.3%) | 74 (39.6%) | | |

* Significant at $p < 0.05$, F = frequency, % = percent.

## Associations of malaria case and knowledge with socio-demographic data

The association between the selected socio-demographic characteristics of the respondents with malaria knowledge was examined using Pearson Chi-square test. As shown in Table 5, sex, age, and family size were not significantly associated. However, livelihood, and education status of the respondents had a significantly association with knowledge of malaria ($\chi2 = 6.219$, $p = 0.013$ and $\chi2 = 8.201$, $p = 0.006$), respectively. Over 70% of the respondents (non-farmers) had good and higher knowledge about malaria. Likewise, the educational status of elementary and above (74.9%) scored good knowledge about malaria, of which 35.3% and 39.6% of them scored good and higher knowledge about malaria, respectively.

Logistic regression model indicated that all predictor variables were not significantly associated with the malaria case except the age (Table 6). Age group of 18–30 years old were about five times more affected by malaria than the other group with AOR = 5.982; 95% CI = 1.715–20.863; p-value = 0.002.

## Discussion

Findings of the present study showed that the positivity rate of malaria, community awareness about cause, transmission, prevention and control of malaria, knowledge about malaria, and use of ITNs as the best control strategies. The ever increasing rate of malaria [9], lack of community awareness towards malaria and inconsistent use of ITNs hinder the malaria prevention and control strategies of the local government. These results suggest that comprehensive interventions must be implemented for effective control of malaria.

This study revealed that the prevalence of malaria in the Maygaba town was found to be 4.7%, this finding is much higher than the prevalence reported in Abeshge (0.25%) [21] and Shewa Robit town (2.8%) [22] south central and northeastern regions of Ethiopia, respectively.

**Table 6.  Logistic regression analysis of factors associated with malaria case (n = 403).**

| Variable | Malaria Case | | *p*-value | AOR (95% CI) |
|---|---|---|---|---|
| | Yes (n = 19) | No (n = 384) | | |
| Sex | | | | |
| Male | 12 | 236 | 1.0 | 1.075 (0.414–2.792) |
| Female | 7 | 148 | | |
| Age | | | | |
| 18–30 | 16 | 181 | 0.002* | 5.982 (1.715–20.863) |
| > 31 | 3 | 203 | | |
| Education Status | | | | |
| Illiterate | 14 | 202 | 0.155 | 2.296 (0.811–6.500) |
| Elementary & above | 5 | 182 | | |
| Livelihood | | | | |
| Farmer | 14 | 205 | 0.155 | 0.436 (0.756–1.233) |
| Non farmer | 5 | 179 | | |
| Family Size | | | | |
| > 5 person | 15 | 236 | 0.222 | 2.176 (0.708–6.684) |
| ≤ 5 person | 4 | 148 | | |
| Livestock | | | | |
| Yes | 12 | 264 | 0.618 | 0.779 (0.299-2-2.029) |
| No | 7 | 120 | | |
| Type of house | | | | |
| Conventional | 7 | 165 | 0.643 | 1.292 (0.498–3.352) |
| Modern | 12 | 219 | | |

*Significant at *p* < 0.05, AOR = Adjust Odds Ratio CI = confidence interval.

However, lower than other regions of Ethiopia such as Dembia (6.7%) [26] and Dejen districts (12.4%) [27] in northwest and east Gojam zones, respectively. Since malaria infection and incidence depend on different factors such as climate, landscape and altitude of a given area, the discrepancies may arise from these listed factors. Furthermore, these differences might also be occurred due to the methodological capability and the type of diagnostic tools used, the established malaria control facilities of the areas, the respondents and social settings.

This study also revealed that malaria parasitaemia among male adults was higher than in females, suggesting higher exposure of males to malaria infection in the study area. This finding is comparable with a similar study reported in other malaria endemic areas of Ethiopia [21]. Because males' had a greater occupational risk of getting the disease than women.

Despite the fact that participants under 18 years were not involved in our study, the detection of malaria parasites in the age group of 18–30 years old was higher compared with those over 31 years old. This finding is in agreement with the studies conducted in adult malaria prevalence surveys in the northwest, south central and east Shewa regions of Ethiopia [21, 26, 28] and also the studies involved all age groups in northwest Tigray and Amhara regional states of Ethiopia [29, 30]. These age groups, particularly the males were usually engaged in outdoor activities including farming and irrigation. However, in some other settings of Ethiopia and Africa, a higher prevalence rate was observed among 5–14 years [29]. These conditions may occur due to socio-economic status, food insecurity, poor housing, health care intervention and social settings [31, 32].

In the current study, *P. falciparum* was predominantly found and became the major (63.2%) contributor of morbidity and mortality in the study area. This is in agreement with

the national report [4, 33] as well as the reports of most other regions of Ethiopia [3, 26]. Malaria is not a year-round phenomenon in Ethiopia, however, there are two main malaria transmission seasons following the two rainy seasons per annum. The existence of parasitaemia in the healthy looking individuals was conducted during the small rainy season. Moreover, the study area is characterized by the presence of sunshine throughout the year [34]. Thus availability of suitable temperature influences the developmental process and survival of the parasite and the mosquito that spreads malaria. Besides temperature, rainfall also helps the breeding sites and increases humidity which in turn creates a conducive environment for survival of adult mosquitoes [22].

The cross-sectional study revealed that the majority (91.3%) of the respondents considered malaria as their major public health problem and over (75%) of the respondents also identified that such disease is transmitted through the biting of mosquito. The awareness of the respondents in this study was slightly lower than other findings elsewhere [21, 22], while the knowledge about malaria transmission in our study group is in agreement with Alelign and Petros [35]. The overall variations between regions and sometimes within regions might be due to lack of home-to-home health extension services.

Besides awareness about the causes and ways of transmission among the respondents, the knowledge of mosquito biting time and prevention measures among the societies are quite important. In this regard, our results indicated that the vast majority (86.6%) of the respondents had knowledge about suitable biting time of mosquito and about 82.4% of the respondents also believed that malaria is preventable. This finding is comparable with the study conducted in Shewa Robit, northeastern Ethiopia [22], but higher than a similar report in another setting [7].

Similarly, about 81.0% of the respondents mentioned the prevention of ITN during sleeping. This result indicates the respondents' knowledge about the cause and transmission method of malaria was quite comparable with other findings elsewhere [20, 21], but lower than the reports in Woreta, northwest Ethiopia [35]. The variation may be due to the extent of malaria in the area and socio-cultural differences found in the indicated settings.

Although several treatment seeking practices were mentioned by the respondents, nonetheless the majority (82.3%) of them preferred health centers for the treatment of malaria. This finding is consistent with the study conducted elsewhere [21]. The respondents' preference might be due to the availability of health centers rather than zonal or referral hospitals in the study area. Thus, providing malaria related facilities in such centers is an important strategy for the reduction of malaria morbidity and mortality in the district.

In addition to assessing the knowledge about malaria vectors, ways of transmission and treatment seeking behaviors of the community, proper knowledge about prevention and control of malaria as well as the vector are also important to reduce the incidence of the disease among the rural community. In this regard, combined KAP and proper ITN utilization among the rural settings are vital since the extent of understanding of the preventive and control measure varies among communities and individual households. Thus, taking ITN as a major malaria prevention tool [36], over three fourth of our study groups had enough information about ITN although their sources were varied. However, only 64.0% owned ITNs, which is consistent with 2016 countrywide coverage of ITNs [3] but lower than other regions of the country such as Benishangul-Gumuz (90.9%) [37], Arsi zone (84.2%) [8] and Gambella region (81.7%) [38]. On the other hand, our finding is higher than the study conducted in Raya Azebo and Harari regions of Ethiopia [7, 15]. This implies that there was variation in the ownerships of ITNs even in the adjacent regions. The low ITNs ownership in this study indicates the absence of well-established health extension service and committed local administrators that give emphasis to the distribution of ITNs in the district.

Additionally, a substantial number (36.2%) of the respondents did not possess ITN due to two main reasons, 56.2%, not received and 43.8%, worn-out. This result indicated that there was an unbalanced and unsustainable distribution of ITN in the district. Therefore, regular distribution of the ITNs by the health extension workers and timely replacements of the worn-out ITNs via malaria campaign are important for effective prevention and control of malaria in the district.

On other hand, the effectiveness of ITNs depends on regular and consistent use, and people's perception towards its utilization [36, 39]. In our study 83.6% of households had a positive attitude towards the benefits of sleeping under ITN and also about 87% of them properly understood its usage. Nevertheless, only 67.3% of them used ITNs last night at the time of interview. This finding is higher than similar studies conducted in Raya Azebo and Arsi zone, Ethiopia [7, 8] but lower than the reports of other settings in Ethiopia [29]. The rest of the respondents did not use ITN during sleeping, this might be due to a negative attitude towards ITN or uncertainty about the use of ITN because of various reasons as mentioned by Taremwa et al [20]. Furthermore, the use of ITNs were associated with several factors such as education level, knowledge of malaria and community engagement, and socio-economic status of the society [16].

In this study we found that 28.4% and 67% of the households used ITN regularly and during peak season of malaria, respectively. Interestingly, priority was given to mothers, children and pregnant women in the family to use ITN. This was a common practice of most communities in Ethiopia [29]. In terms of ITNs utilization experience, almost all the households had been using their ITNs at least for one year and above. This finding is in line with the findings of other studies elsewhere [40]. ITN is not equivalent to LLIN, therefore, it requires re-treatment with insecticide [36], while only a quarter of them re-treated their ITNs in the period of two to three years of ownership. Thus, the trend of re-treatment of ITNs with insecticide and repair of teared INTs has to be encouraged among the society [16, 41].

In general the findings of this study may help to suggest possible solutions to most African and other malaria endemic regions of the world by providing first-hand information about adult's malaria positivity and community awareness towards the cause, transmission, prevention and control of malaria and use of ITNs. Furthermore, it would be important for policy makers and malaria supporting partners to take appropriate measures for the effective prevention and control of malaria in developing nations including Ethiopia.

Limitations of this study includes failure to use advanced diagnostic methods such as molecular diagnosis using polymerase chain reaction (PCR) and rapid diagnostic test (RDT) to further confirm the results obtained through microscopic examination. Additionally, it could have been better to design the cross-sectional survey accompanied with direct observation of ITNs usage in the study area. Besides lack of direct observation, only one person per household was included in the assessment of ITNs usage due to the difficulty of obtaining the whole family at home during the survey. Finally, the absence of prevalence data from children may affect the prevalence rate of malaria in the district.

## Conclusions

In conclusion, the community based parasitological survey discovered nineteen slide-positive malaria cases. Majority of the respondents were aware of the cause, communicability, preventability and curability of the disease. ITNs use was identified as the main malaria prevention method. Although the community's KAP and ITNs possession rate is acceptable, this does not elucidate substantial usage, sustainable distribution and timely replacement of old ITNs in the study area. This negatively influenced the prevention and control effort of malaria among the

predisposed societies of the locality. Therefore, comprehensive malaria intervention measures should be in place for the effective control of malaria in the town. These could be full coverage of ITNs, regular monitoring and replacements of old and worn-out ITNs and educate the community about usage and re-treatments of the possessed ITNs are expected from the local government and all stakeholders.

## Supporting information

**S1 Text. Questionnaire used for household survey in this study.**
(DOC)

## Acknowledgments

We gratefully acknowledge Addis Ababa University, School of Graduate Program for the provision of technical and administrative support to this project. We also thank Dr. Asfaw Degu for his critical revision and correction of this manuscript. The authors' also would like to thank College of Natural and Computational Sciences, Institutional Review Board (CNS-IRB) for ethical approval. Authors are also thankful to Federal Minister of Education School of Biological Sciences and Biotechnology, MHC and laboratory technicians and study participants' for their valuable information and support.

## Author Contributions

**Conceptualization:** Tsegay Gebremaryam Yhdego, Asnake Desalegn Gardew, Fitsum Tigu Yifat.

**Data curation:** Tsegay Gebremaryam Yhdego.

**Formal analysis:** Tsegay Gebremaryam Yhdego, Asnake Desalegn Gardew, Fitsum Tigu Yifat.

**Investigation:** Tsegay Gebremaryam Yhdego, Fitsum Tigu Yifat.

**Methodology:** Tsegay Gebremaryam Yhdego, Asnake Desalegn Gardew, Fitsum Tigu Yifat.

**Project administration:** Fitsum Tigu Yifat.

**Resources:** Fitsum Tigu Yifat.

**Software:** Asnake Desalegn Gardew, Fitsum Tigu Yifat.

**Supervision:** Fitsum Tigu Yifat.

**Writing – original draft:** Fitsum Tigu Yifat.

**Writing – review & editing:** Asnake Desalegn Gardew, Fitsum Tigu Yifat.

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
