## [Decision Letter · Decision Letter 0]

6 Aug 2021

 PGPH-D-21-00037 The prevalence of asymptomatic malaria parasitemia and associated knowledge, attitudes and practices and the use of insecticide-treated mosquito net in a rural setting in Ethiopia PLOS Global Public Health

Dear Dr. Fitsum Tigu Yifat,

Thank you for submitting your manuscript to PLOS Global Public Health. After careful consideration, we feel that it has merit but does not fully meet PLOS Global Public Health’s publication criteria as it currently stands. Therefore, we invite you to submit a revised version of the manuscript that addresses the points raised during the review process.

We look forward to receiving your revised manuscript.

Kind regards,

David Musoke, PhD

Academic Editor

Journal Requirements:

Additional Editor Comments (if provided):

Reviewers' comments:

Reviewer's Responses to Questions

**Comments to the Author**

1. Does this manuscript meet PLOS Global Public Health’s publication criteria? Is the manuscript technically sound, and do the data support the conclusions? The manuscript must describe methodologically and ethically rigorous research with conclusions that are appropriately drawn based on the data presented.

Reviewer #1: Partly

Reviewer #2: Yes

2. Has the statistical analysis been performed appropriately and rigorously?

Reviewer #1: No

Reviewer #2: Yes

3. Have the authors made all data underlying the findings in their manuscript fully available (please refer to the Data Availability Statement at the start of the manuscript PDF file)?

Reviewer #1: Yes

Reviewer #2: No

4. Is the manuscript presented in an intelligible fashion and written in standard English?

Reviewer #1: Yes

Reviewer #2: Yes

5. Review Comments to the Author

Reviewer #1: Review of manuscript PGPH-D-21-00037

The prevalence of asymptomatic malaria parasitemia and associated knowledge, attitudes and practices and the use of insecticide-treated mosquito net in a rural setting in Ethiopia

This paper addresses an important question on malaria prevalence among adults and use of nets for malaria control in Ethiopia. By looking at malaria burden, we are better equipped to assess the short and long term impacts of various control strategies including use of mosquito nets.

Title

The title of the study is long and the key outcome of interest is not clear. The study population of interest is not mentioned in the title. The specific study area should be mentioned as well. The study title could be rephrased to read “Malaria prevalence, insecticide-treated mosquito net use and associated factors among household heads in Maygaba town, Ethiopia”

Abstract

The introduction should highlight key issues on malaria prevalence, use of ITNs and associated factors. The study aim should align with the suggested study title. Include information on study population, sampling, and key outcome measures in the study methodology. There should be results for key study objectives that should be aligned with the proposed study title. When describing associations, show how it influences the outcome of interest. The conclusion and recommendation should be rewritten to focus on the key outcomes of interest. It is not true that malaria prevalence was high (4.7%) in the study population.

Introduction.

The introduction is very long and not well focused on the title of the study. The authors have largely dwelt on the global and local malaria burden and control measures. It would be helpful to outline why the problem or question is important. Focus on what is known about malaria burden and use of insecticide-treated mosquito net for malaria control and associated factors among household heads. Recent literature on what is known about prevalence and factors associated with use of insecticide-treated mosquito nets should be included in the introduction. The authors should state the relevance of the research to other research that has been done. Please include contextual information on what is being done to control malaria in the study area.

Materials and methods.

Study design, area and population.

The study design and area is well described. The authors should give a strong justification for carrying out the malaria prevalence study among adults and the utility of such information. Malaria burden among adults is not a true reflection of malaria burden in the community because adults have partial immunity to malaria in endemic areas. Malaria prevalence studies are routinely done in children because they are not immune to malaria and carry the highest burden of malaria. Please justify why information on net use was collected for one person in a household. Information on use of bednets is best collected from all household members in a selected house hold as per the Roll Back Malaria (RBM) indicators. The population inclusion and exclusion criteria should be well stated including consent to participate in the study.

Sample size and sampling

It is not clear what key outcome measure was used for sample size estimation. The authors should use one key outcome to estimate sample size. The value of “p” was assumed as 0.5 yet there is existing literature from which values can be got, please clarify why a value from literature was not used. The authors state that they used a multi-stage cluster random sampling technique. To adjust for the variations between the clusters, the authors should have adjusted sample size by a design effect and possibly used a different formula for sample size estimation, please clarify why this was not the case. The authors state that 3 villages (Korarit,

Maygaba and Adijamus) with a total population of 16,781 and 3,814 households were sampled out of the 6 villages in the study area. Please state the criteria for selecting only 3 villages and how sampling was done for the 3 out of the 6 villages in the study area. Another confusing tier for sampling is the “Gujiles”. The authors should clearly distinguish between a Gujile and a village and how they interrelate. Clearly outline how households were selected from the “villages/Gujiles”. Systematic random sampling requires a list of households which has not been mentioned by the authors. The ideal sampling criterion should have been sampling proportionate to size. The authors should further describe how the respondents were sampled from each household.

Data collection and analysis

The authors should describe the key outcome measures and for each give a detailed description of how the outcome was measured. The authors have coined a composite index they refer to as “knowledge about malaria and ITN utilization”. Any composite index used should be a meaningful index. The authors state that questionnaires were pre-tested by preliminary survey in some Gujiles, please clarify if these were outside the sampled areas. Describe in detail how quality control of smear reading was done and how discrepant readings were resolved. In the data analysis please describe how you addressed confounding.

Results.

The results section should be re written. Results should be presented per objective. Results for key outcomes such as parasite prevalence, awareness about cause, transmission, control and prevention of malaria, knowledge about malaria, and use of ITNs should initially be presented separately. A bivariate analysis of the dependent and independent variables can then be made thereafter. To adjust for confounding, a multivariable analysis should be done possibly using odds ratios or prevalence ratios for the analysis. The authors should use the recommended RBM indicators to describe net use. Please note that history of fever or temperature measurement were not taken on study participants so you in effect did not determine prevalence of asymptomatic malaria parasitaemia.

Discussion

The discussion is long but not focused on study objectives. The discussion should focus primarily on results from all the study objectives. The authors should discuss the significance of all key findings before relating to other researched work. The authors should discuss if results from the study can be generalized to other areas of the same setting. Study limitations should be comprehensively discussed including the limitations in the study design. Comment on whether results can be generalised to other settings in Africa and across the world. There should be a conclusion for every objective and recommendations should be based on the study finding.

General comments

The paper needs major revisions, corrections are needed for grammatical errors, and for improving the content, organization and flow of the write up.

Reviewer #2: All comments and suggestions uploaded; I think, in this study, what was actually measured was positivity rate but not prevalence. I would strongly recommend a change. Otherwise, if it is a prevalence, let the try to do weighted analysis. Again, in the title, the term “practices: and use more or less mean the same. May think about dropping use and remain with the KAP.

6. PLOS authors have the option to publish the peer review history of their article (what does this mean?). If published, this will include your full peer review and any attached files.

**Do you want your identity to be public for this peer review?** For information about this choice, including consent withdrawal, please see our Privacy Policy.

Reviewer #1: No

Reviewer #2: **Yes: **Simon Kasasa, PhD

---

## [Decision Letter · Decision Letter 1]

8 Oct 2021

PGPH-D-21-00037R1

Malaria prevalence, insecticide-treated mosquito net use and associated factors among household heads in Maygaba town, Ethiopia

Dear Dr. Yifat,

Thank you for submitting your manuscript to PLOS Global Public Health. After careful consideration, we feel that it has merit but does not fully meet PLOS Global Public Health’s publication criteria as it currently stands. Therefore, we invite you to submit a revised version of the manuscript that addresses the points raised during the review process.

Please submit your revised manuscript by 29 October 2021. If you will need more time than this to complete your revisions, please reply to this message or contact the journal office at globalpubhealth@plos.org. Please include the following items when submitting your revised manuscript:

We look forward to receiving your revised manuscript.

Kind regards,

David Musoke, PhD

Academic Editor

Journal Requirements:

Additional Editor Comments (if provided):

May you revise the manuscript based on the latest comments, paying particularly attention to thorough English editing.

Reviewers' comments:

Reviewer's Responses to Questions

**Comments to the Author**

1. If the authors have adequately addressed your comments raised in a previous round of review and you feel that this manuscript is now acceptable for publication, you may indicate that here to bypass the “Comments to the Author” section, enter your conflict of interest statement in the “Confidential to Editor” section, and submit your "Accept" recommendation.

Reviewer #1: (No Response)

2. Does this manuscript meet PLOS Global Public Health’s publication criteria? Is the manuscript technically sound, and do the data support the conclusions? The manuscript must describe methodologically and ethically rigorous research with conclusions that are appropriately drawn based on the data presented.

Reviewer #1: Partly

3. Has the statistical analysis been performed appropriately and rigorously?

Reviewer #1: No

4. Have the authors made all data underlying the findings in their manuscript fully available (please refer to the Data Availability Statement at the start of the manuscript PDF file)?

Reviewer #1: Yes

5. Is the manuscript presented in an intelligible fashion and written in standard English?

Reviewer #1: Yes

6. Review Comments to the Author

Reviewer #1: (No Response)

7. PLOS authors have the option to publish the peer review history of their article (what does this mean?). If published, this will include your full peer review and any attached files.

**Do you want your identity to be public for this peer review?** For information about this choice, including consent withdrawal, please see our Privacy Policy.

Reviewer #1: **Yes: **Adoke Yeka

---

## [Decision Letter · Decision Letter 2]

23 Nov 2021

PGPH-D-21-00037R2

Malaria prevalence, insecticide-treated mosquito net use and associated factors among household heads in Maygaba town, Ethiopia

Dear Dr. Yifat,

Thank you for submitting your manuscript to PLOS Global Public Health. After careful consideration, we feel that it has merit but does not fully meet PLOS Global Public Health’s publication criteria as it currently stands. Therefore, we invite you to submit a revised version of the manuscript that addresses the points raised during the review process.

We look forward to receiving your revised manuscript.

Kind regards,

David Musoke, PhD

Academic Editor

Journal Requirements:

Additional Editor Comments (if provided):

Kindly take note of the latest comments especially regarding measurement of the outcome variables. In addition, thorough proof-reading of the entire manuscript is needed. The authors may want to get external support on this.

Reviewers' comments:

Reviewer's Responses to Questions

**Comments to the Author**

1. If the authors have adequately addressed your comments raised in a previous round of review and you feel that this manuscript is now acceptable for publication, you may indicate that here to bypass the “Comments to the Author” section, enter your conflict of interest statement in the “Confidential to Editor” section, and submit your "Accept" recommendation.

Reviewer #1: (No Response)

2. Does this manuscript meet PLOS Global Public Health’s publication criteria? Is the manuscript technically sound, and do the data support the conclusions? The manuscript must describe methodologically and ethically rigorous research with conclusions that are appropriately drawn based on the data presented.

Reviewer #1: Yes

3. Has the statistical analysis been performed appropriately and rigorously?

Reviewer #1: Yes

4. Have the authors made all data underlying the findings in their manuscript fully available (please refer to the Data Availability Statement at the start of the manuscript PDF file)?

Reviewer #1: Yes

5. Is the manuscript presented in an intelligible fashion and written in standard English?

Reviewer #1: No

6. Review Comments to the Author

Reviewer #1: • The authors should mention what the key outcome measure(s) were in the methods section and describe how composite outcomes were measured.

• The section on KAP scoring system needs to be made clearer. The authors stated that a KAP scoring system was adopted from a previously published method. Briefly mention what was being assessed under this system. The authors should include results for the KAP scores in the paper, if not the whole section should be deleted.

• The authors have shown results of association of malaria knowledge with socio-demographic data. Please describe how the outcome “malaria knowledge” was measured.

• Please include results for factors associated with net use and discuss these findings in the discussion

• There are still many grammatical errors in the paper, the paper may benefit from a language edit.

7. PLOS authors have the option to publish the peer review history of their article (what does this mean?). If published, this will include your full peer review and any attached files.

**Do you want your identity to be public for this peer review?** For information about this choice, including consent withdrawal, please see our Privacy Policy.

Reviewer #1: **Yes: **Adoke Yeka

---

## [Editor Report · Decision Letter 3]

2 Mar 2022

Malaria prevalence, knowledge and associated factors among household heads in Maygaba town, Ethiopia

PGPH-D-21-00037R3

Dear Yifat,

We are pleased to inform you that your manuscript 'Malaria prevalence, knowledge and associated factors among household heads in Maygaba town, Ethiopia' has been provisionally accepted for publication in PLOS Global Public Health.

Best regards,

David Musoke, PhD

Academic Editor

Congratulations on getting your research accepted for publication.